# New 6-Minute-Walking Test Parameter—Distance/Desaturation Index (DDI) Correctly Diagnoses Short-Term Response to Immunomodulatory Therapy in Hypersensitivity Pneumonitis

**DOI:** 10.3390/diagnostics13061109

**Published:** 2023-03-15

**Authors:** Katarzyna B. Lewandowska, Małgorzata Sobiecka, Piotr W. Boros, Małgorzata Dybowska, Inga Barańska, Małgorzata E. Jędrych, Agata Gładzka, Witold Z. Tomkowski, Monika Szturmowicz

**Affiliations:** 11st Department of Lung Diseases, National Research Institute of Tuberculosis and Lung Diseases, 01-138 Warsaw, Poland; 2Department of Respiratory Physiopathology, National Research Institute of Tuberculosis and Lung Diseases, 01-138 Warsaw, Poland; 3Department of Radiology, National Research Institute of Tuberculosis and Lung Diseases, 01-138 Warsaw, Poland; 4Department of Respiratory Rehabilitation, National Research Institute of Tuberculosis and Lung Diseases, 01-138 Warsaw, Poland

**Keywords:** hypersensitivity pneumonitis, 6-minute walk test, 6-minute walk distance, desaturation, distance-saturation product, distance/desaturation index

## Abstract

The six-minute-walking test (6MWT) is an easy-to-perform, cheap and valuable tool to assess the physical performance of patients. It has been used as one of the endpoints in many clinical trials investigating treatment efficacy in pulmonary arterial hypertension and idiopathic pulmonary fibrosis. However, the utility of 6MWT in patients diagnosed with hypersensitivity pneumonitis (HP) is still under investigation. The aim of the present retrospective study was to assess the value of different 6MWT parameters, including the newly developed distance-desaturation index (DDI), to evaluate immunomodulatory treatment outcomes in HP patients. Methods: 6MWT parameters (distance, initial saturation, final saturation, desaturation, distance-saturation product (DSP), and DDI) were analyzed at baseline and after 3 to 6 months of treatment with corticosteroids alone or in combination with azathioprine. Results: 91 consecutive HP patients diagnosed and treated in a single pulmonary unit from 2005 to 2017 entered the study. There were 44 (48%) males and 52 (57%) patients with fibrotic HP (fHP). Sixty-three patients (69%) responded to treatment (responders) and 28 (31%) did not respond (non-responders). In the responders group, all parameters assessed during 6MWT significantly improved, whereas in non-responders, they worsened. Medians (95% CI) of best indices were post-treatment DDI/baseline DDI—1.67 (1.85–3.63) in responders versus 0.88 (0.7–1.73) in non-responders (*p* = 0.0001) and change in walking distance–51 m (36–72 m) in responders, versus 10.5 m (−61.2–27.9) in non-responders (*p* = 0.0056). The area under the curve (AUC) of receiver operating characteristics (ROC) for post-treatment DDI/baseline DDI was 0.74 and the optimal cut-off was 1.075, with 71% of specificity and 71% of sensitivity. Conclusions: 6MWT may be used as a tool to assess and monitor the response to immunomodulatory therapy in HP patients, especially if indices incorporating both distance and desaturation are used. Based on the present study results, we recommend 6MWD and DDI use, in addition to FVC and TL,co, to monitor treatment efficacy in patients with interstitial lung diseases.

## 1. Introduction 

The six-minute-walking test (6MWT) is a valuable tool to assess exercise capacity in patients with various respiratory and circulatory diseases. It was primarily developed to diagnose physical compromise in patients with chronic obstructive pulmonary disease (COPD), and later its use has been extended to other pulmonary and cardiovascular disorders. 6MWT reflects the patient’s mobility, exercise tolerance, and gas exchange efficacy in the most natural and physiological circumstances, i.e. self-paced walking. 6MWT parameters are related to different patient characteristics, including age, sex, height, baseline physical activity, muscle strength, frailty, neurological abnormalities, etc. [1]. The test is performed on a flat corridor of thirty meters in length, and the patient is instructed to walk as quickly as possible to cover the longest possible distance [1]. The oxygen saturation during the test is measured with the pulse-oximeter, and the levels of dyspnea and fatigue are indicated on the Borg scale, where a score of 0 means no dyspnea or fatigue, and a score of 10 is the worst possible dyspnea or fatigue [1]. Due to its simplicity, 6MWT is often used in patients with cardiovascular disorders as well as in those with interstitial lung diseases (ILDs) to diagnose basic physical compromise and to monitor the disease course [2,3,4].

In patients with pulmonary arterial hypertension (PAH), 6-min walking distance (6MWD) is used in an algorithm to prediction baseline mortality risk [5]. 6MWD also serves as an important indicator of PAH treatment effectiveness [5]. Clinical application of 6MWT also concerns patients with other types of pulmonary hypertension. Sobiecka et al. found that the composite model based on age, 6MWT, and TLC/TLCO was highly specific for predicting pulmonary hypertension on echocardiography in patients with various ILDs [6]. A recently published INCREASE trial, which concerned the efficacy of inhaled treprostinil in patients with pulmonary hypertension due to various ILDs, documented a significant 6MWD increase in the treprostinil arm [7].

6MWT is also used in the clinical assessment of patients with ILDs, mainly idiopathic pulmonary fibrosis (IPF) and connective tissue disease-related ILDs (CTD-ILD) [3,8,9]. In IPF patients, 6MWD has been widely used as a secondary endpoint in randomized clinical trials (RCT) involving various types of therapy [10,11,12,13]. The data on the clinical utility of 6MWT in patients with hypersensitivity pneumonitis (HP) is scarce. Two RCTs concerning the efficacy of anti-fibrotic treatment enrolled patients with HP, i.e., the INBUILD trial and the RELIEF trial [14,15]. In the INBUILD trial, the 6MWT parameters were not evaluated [14]. In the RELIEF trial, the difference in 6MWD before and after treatment with pirfenidone was assessed as a secondary end-point [15]. 

As HP patients are usually younger than those with IPF, and they have a better life expectancy [16,17,18], 6MWT performance may differ from that documented in IPF patients. Thus, the aim of the present retrospective study was to assess the utility of different 6MWT parameters including the newly developed distance-desaturation index (DDI) in the evaluation of immunomodulatory treatment outcome in HP patients. 

## 2. Materials and Methods

### 2.1. Diagnosis of HP and Treatment Effect Assessment 

We retrospectively analyzed baseline and post-treatment 6MWT parameters in consecutive HP patients diagnosed in the 1st Department of Lung Diseases, National Research Institute of Tuberculosis and Lung Diseases, Warsaw, Poland, between 2005 and 2017, who received immunomodulatory therapy. The principles of the HP diagnostic algorithm used in our center were published previously [18,19]. In short, the diagnosis of HP was based on clinical, radiological, and/or histopathological features, i.e., known exposures to organic dust, characteristic changes in high-resolution computed tomography (HRCT) of the lungs, and increased lymphocyte percentage in broncho-alveolar lavage fluid (BALF). In cases of diagnostic difficulty, lung biopsies were performed. The diagnostic approach was in line with the recently published ATS/JRS/ALAT guidelines [20].

Immunomodulatory therapy consisted of prednisone alone (0.5 mg/kg/day) or in combination with azathioprine (100–150 mg/day). The treatment was implemented at the discretion of clinicians after assessing the patient’s functional status, including pulmonary function tests (PFTs), chest imaging, and 6MWT, and taking into consideration chances for antigen exposure avoidance on a case-by-case basis. Response to treatment was assessed with PFTs and chest X-ray examination after 3–6 months of therapy. A patient was considered a responder if either VCmax or TL,co after treatment improved (i.e. increased by any value), and the changes in the X-ray image were less visible or stable compared to the baseline. On the other hand, if the VCmax or TL,co worsened (i.e. decreased by any value) and the changes in the X-ray were stable or more pronounced, the patient was considered a non-responder. A description of the categories of treatment response is presented in Table 1.

6MWT was performed as a routine procedure in every patient who was able to walk on a flat 30 m long corridor, with baseline and sixth-minute room air oxygen saturation, according to ATS and Polish Respiratory Society guidelines [1,21]. We analyzed different parameters measured during 6MWT separately (distance, initial saturation (SpO_2_ 1), end-test saturation (SpO_2_ 2), and desaturation) and also their combinations: distance-saturation product (DSP), which was introduced by Lettieri et al. [22], and a new one, the distance/desaturation index (DDI). DDI was calculated by dividing the distance walked, measured in meters, by the difference between initial and final saturation during the test. In the case of no saturation change, we used “1” as a numerator instead of “0” to maintain the correctness of the mathematical operation. For example, if the patient walked 400 m, SpO_2_ 1 was 96%, and SpO_2_ 2 was 88%, the DDI would be calculated as follows: 400/(96–88) = 400/8 = 50. We presumed DDI would be a more sensitive parameter in the assessment of the patient’s physical performance than single parameters because it incorporates the magnitude of desaturation, which is more informative than the lowest saturation measured. For example, the lowest saturation of 90% may be a result of a substantial drop from the initial 99% or insignificant if the patient’s initial saturation was 91%. 

### 2.2. Statistical Analysis

Statistical analysis was performed using GraphPad Prism 9.4.1 (458) on 18 July 2022 (GraphPad Software, LCC, San Diego, CA, USA). *p* values of <0.05 were considered statistically significant. The values were presented as means ± SD or median and CI. Between-group comparison for continuous variables in two groups was assessed with the T-Student test, Fisher exact test, Mann-Whitney test, or Wilcoxon test, where appropriate. Correlations between different parameters were assessed using the Spearman correlation test. Cut-off values were established with the use of receiver operating characteristics (ROC) curves.

### 2.3. Regulatory Board Approval

The study was accepted by the Institutional Ethics Committee of the National Research Institute of Tuberculosis and Lung Diseases in Warsaw, Poland (KB-14/2019). Patients’ consent was waived by the Ethics Committee because of the retrospective character of the study.

## 3. Results

### Baseline Characteristics of the Study Group

91 HP patients entered the study: 44 males (48%) and 47 females (52%), with a mean age of 51.6 years (+/− 10.32 years). Fibrotic HP (fHP) was recognized in 52 patients (57%), and non-fibrotic HP (non-fHP) in 39 (43%). Detailed characteristics of the study group are presented in Table 2.

The comparison of baseline 6MWT parameters in fibrotic and non-fibrotic HP is presented in Table 3. No significant differences were noted, apart from the higher degree of desaturation during the 6MWT test in the fHP patients compared to non-fHP (borderline significance). Therefore, we decided to analyze both groups as a whole.

A correlation between baseline 6MWT and PFT parameters is presented in Table 4. Baseline desaturation during 6MWT correlated negatively with both VC max %pred and TL,co %pred. Baseline DDI correlated positively with both VC max %pred and TL,co %pred, whereas 6MWD and DSP correlated with TL,co %pred only.

Based on the results of PFT and chest x-rays, according to the above-described criteria, 63 patients (69%) were defined as responders and 28 (31%) as non-responders to immunomodulatory therapy. The analysis of 6MWT results before and after therapy in the whole group showed significant improvement of all the analyzed 6MWT parameters, except for DDI. Nevertheless, after separating responders from non-responders, in responders the improvement remained significant for all parameters including DDI, whereas, in non-responders, only 6MWD and DSP differed significantly (Table 5). The new 6MWT parameter, DDI, improved more than two-fold in responders and worsened insignificantly in non-responders.

Based on the results obtained, the significance of treatment-related changes in 6MWT parameters in responders compared to non-responders was analyzed (Table 6).

The analysis of post-treatment to pre-treatment DDI score revealed that in responders, the median value was 1.67 (95%CI: 1.85 to 3.63), and in non-responders it was 0.88 (95%CI: 0.7 to 1.73) (*p* = 0.0001). The difference in post-treatment vs. pre-treatment walking distance also significantly differentiated responders from non-responders, whereas the changes in desaturation range were insignificant. Furthermore, a comparison of post-treatment to pre-treatment DSP score according to treatment response showed that the median was 1.09 in both responders and non-responders (*p* = 0.6428).

The value of DDI post/DDI pre in diagnosing the response to therapy is presented on the ROC curve (Figure 1). ROC AUC was 0.74 (95%CI: 0.6314–0.8578).

DDI post/DDI pre higher than 1.075 separated responders from non-responders with 71% sensitivity and specificity, 0.85 positive predictive value (PPV), 0.53 negative predictive value (NPV), and 6.25 odds ratio (95% CI: 2.22–17.32). For the remaining parameters, ROC AUC were 0.68 (95% CI: 0.5560–0.8602) for distance change, 0.5794 (95% CI: 0.4539–0.7049) for desaturation change, and 0.53 (95% CI: 0.3971–0.6646) for DSPpost/DSPpre. Thus, the new parameter, DDI post/DDI pre ratio, showed the best performance in predicting positive response to treatment in our cohort of patients. 

## 4. Discussion

We analyzed the performance of 6MWT for assessing immunomodulatory treatment efficacy in patients with HP. Single 6MWT parameters were used, as well as the combined assessment of distance and desaturation, DSPs and new parameter, DDI. No significant differences were found between the baseline 6MWT results in patients with fibrotic HP compared to non-fibrotic HP; thus, both groups were combined in further analysis. 

The baseline characteristics of the presented study group regarding 6MWT parameters were similar to the ILD patients described by Li et al. [23]. On the other hand, even the fibrotic HP patients from our cohort had longer baseline 6MWT walking distance than the IPF patients from clinical trials, probably due to the lower age of our population [24]. The comparison of 6MWT parameters in different cohorts of ILD patients published in the literature and our cohort is shown in Table 7.

In the present study group, 6MWT desaturation and DDI correlated with VC max %pred and TL,co %pred, whereas 6MWD and DSP correlated with TL,co %pred only. Our findings differ slightly from other published data on ILD patients. In the study of Alhamad et al. in patients with sarcoidosis, 6MWD and DSP correlated with FEV1% pred, FVC % pred, and TLC %pred, but the correlation with TL,co %pred was not assessed [25]. Seema et al. presented significant correlations between 6MWD, pre-test and post-test Borg dyspnea index and spirometry parameters as well as TL,co %pred. [8]. In our opinion, assessment of desaturation itself, or the combination of distance walked and desaturation observed, reflected better pulmonary function impairment as they correlated with both VC max and TL,co.

An increase of 6MWD, as well as DSP and a less pronounced desaturation, was documented after treatment implementation in the entire cohort of our patients. Nevertheless, significant improvement in the majority of 6MWT parameters concerned only the responders. On the other hand, 6MWD and DSP also increased in the non-responders. The only paper concerning treatment-related changes of 6MWT parameters in HP was published by Tony et al., who found a significant increase in 6MWD in non-fibrotic HP patients after eight weeks of treatment with methylprednisolone [26]. The increase of 6MWD after therapy is not always combined with less desaturation. In patients with chronic ILDs, a high tolerance of lower oxygen saturation may be observed. Therefore, patients tend to walk longer distances, despite substantial drops in oxygen saturation.

The desaturation magnitude during walking was rarely analyzed as a separate parameter. A small group of IPF patients was observed in the study of Gupta et al., who considered 10% desaturation in the initial 6MWT as a bad prognostic factor [27]. Lama et al., in their retrospective study on patients with histopathological diagnosis of UIP and NSIP, showed a saturation drop during the 6MWT to be an independent mortality risk factor [28]. In our cohort, a significant decrease in desaturation magnitude was noted after treatment in the responders group, although when the change of desaturation pre- and post-treatment was analyzed, the difference between groups was not significant.

**Table 7 diagnostics-13-01109-t007:** Baseline 6MWT results (means ± SD) in different ILDs (reference-based).

ILD-Type	Ref Nº	6MWD (m)	6MWT SpO_2_-1 (%)	6MWT SpO_2_-2 (%)	6MWT SpO_2_ Desaturation (%)	DSP
IPF	[2]	240 ± 300	95±2	89 ± 7	7±6	-
IPF	[4]	392.4 ± 108.5	-	-	-	-
CTD-ILD	[8]	352.26 ± 57.36	-	-	-	-
PPF	[15]	357.7 ± 99.2	-	-	-	-
IPF	[22]	406.9 ± 71.6	97.4 ± 1.2	89.4 ± 3	7.9 ± 2.8	364.8 ± 67.2
Different ILDs	[23]	457.28 ± 98.4	95.5 ± 2.23	90.5 ± 6.43	4.96 ± 5.57	-
Sarcoidosis	[25]	349 ± 72	95.5 ± 2.4	91.4 ± 7.3	-	320 ± 75
fHP	[26]	232.22 ± 49.53	89.67 ± 3.86	80.56 ± 6.20	-	-
IPF	[28]	355.4 ± 108.2	95.6 ± 1.0	-	7.1 ± 4.1	-
HP	Present study	485.7 ± 106.3	95.38 ± 2.72	87.59 ± 7.38	7.82 ± 6.67	424.2 ± 110.4

ILD—interstitial lung disease, 6MWT—6-min walk test, 6MWD—6 min walk distance, 6MWT SpO_2_-1—6MWT oxygen saturation before the test, 6MWT SpO_2_-2—6MWT oxygen saturation after the test, DSP—distance-saturation product, IPF—idiopathic pulmonary fibrosis, CTD-ILD—connective tissue disease-related ILD, PPF—progressive pulmonary fibrosis, fHP—fibrotic hypersensitivity pneumonitis.

Many doubts concern 6MWD as an endpoint in clinical trials regarding ILDs. The efficacy of treatment may be defined as the absolute or relative change in walking distance. In the randomized clinical trials on pirfenidone in patients with IPF (CAPACITY and ASCEND), the drop of 6MWD by 50 m or more was a component of the composite progression-free survival outcome. The treatment with pirfenidone decreased the proportion of patients experiencing such a decline in 6MWD, or death, by 28% [24]. In the RELIEF trial, where patients with HP comprised 47% of the whole study group, 6MWD decreased more in patients receiving placebo than in those receiving pirfenidone, but the difference was not statistically significant [15].

In our cohort, patients who responded well to immunomodulatory therapy increased their walking distance by more than 50 m. Tony et al. documented that the 6MWD increase from 25 to 44 m, and the magnitude of saturation drop of 2 to 5%, were the indicators of positive treatment effect in non-fibrotic HP patients [26]. Nathan et al. found that the minimal clinically significant change in 6MWD in IPF patients was 21–37 m [29]. We did not define a minimal clinically significant distance change representative for good treatment response. However, based on the presented data, we documented that the mean 6MWD gain in responders compared to non-responders was 40 m.

We speculated that the two calculated indices, DDI and DSP, could be more predictive of treatment response than single 6MWT parameters. DSP was developed by Lettieri et al. to further increase the sensitivity of 6MWT in predicting mortality by combining both the distance walked and the lowest saturation during the test [22]. It performed better than any single 6MWT parameter predicting mortality in IPF patients. DSP was also analyzed by Alhamad et al. in patients with sarcoidosis [25]. In the present study, we checked the utility of DSP and also a slightly different parameter: the distance/desaturation index. This new index was better than DSP in predicting positive treatment effect in our cohort of patients with HP. It more precisely reflected the disabled exercise capacity, considering the difference between baseline and end-walking saturation, and enabled more precise differentiation between responders and non-responders compared to 6MWD. DDI also performed better than DSP and any other 6MWT parameter in predicting a positive response to treatment (ROC AUC 0.74). In our opinion, the DDIpost/DDIpre ratio may serve as a component of composite end-points in clinical trials, but further prospective studies are needed to establish the minimal clinically significant DDI change corresponding to treatment response in different groups of ILD patients, including HP, IPF and other types of progressive pulmonary fibrosis.

### Study Limitations

The most important limitation of our study is its retrospective nature. Nevertheless, we included in the study group consecutive patients with HP, diagnosed according to the same algorithm, who received immunomodulatory therapy due to a significant reduction of lung volumes and/or TL,co. The therapy monitoring included chest X-ray, body plethysmography with TL,co, and 6MWT, performed at each clinical checkpoint. Thus, in our opinion, the presented data were not influenced by their retrospective nature. Secondly, we did not include the control group to observe the differences in 6MWT in non-treated patients. Nevertheless, the aim of the study was to investigate whether, in HP patients receiving immunomodulatory therapy, 6MWT may be applied as an additional tool to assess therapy response, and our results confirmed this hypothesis. Third, we measured only start- and end-test saturation, although the saturation nadir may occur at any time during the test. This was due to the organization of our rehabilitation department and limitations in the access to continuous pulse oximetry during the test.

Despite that, the presented data may increase the confidence in using 6MWT parameters, including the new one, DDI, which performed best in differentiating responders from non-responders, to assess the treatment response in different ILD patients, especially with HP.

## 5. Conclusions

Based on the analysis we performed in the cohort of HP patients, we consider 6MWT to be a valuable clinical tool to assess the treatment effect in those patients. Both distance walked and saturation drop should be taken into consideration when assessing the improvement or deterioration. The newly generated index, DDI, especially if the DDIpost/DDI-pre ratio was calculated, had the best characteristics in therapy monitoring.

## Figures and Tables

**Figure 1 diagnostics-13-01109-f001:**
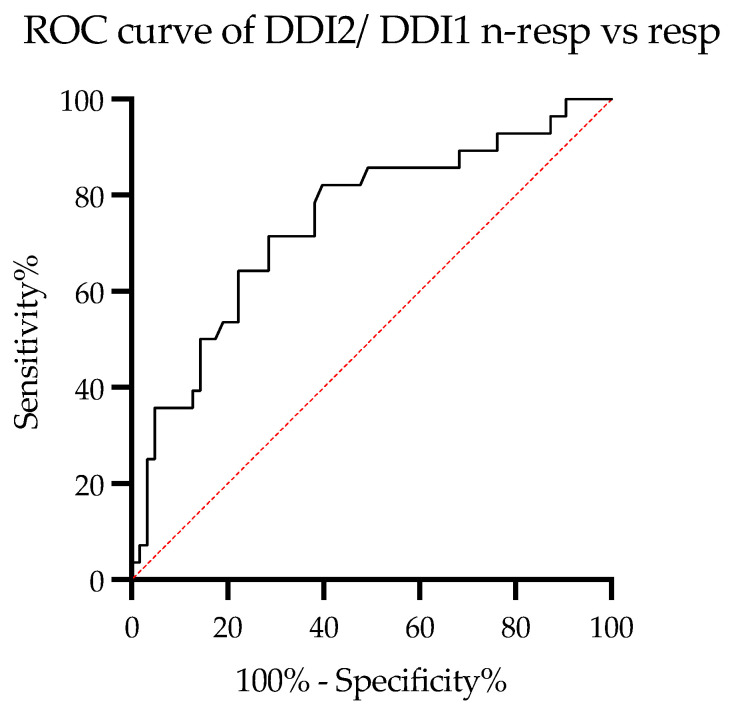
Response to treatment based on DDI post/ DDI pre (ROC curve). DDI2—DDI post, DDI1—DDIpre; n-resp—nonresponders, resp—responders.

**Table 1 diagnostics-13-01109-t001:** Therapy outcome categorization based on PFT and chest X-ray results.

Parameter	Responders	Non-Responders
VCmax or TL,co	Any increase or stable	Any decrease
	AND	AND
Chest X-ray	Stabilization or improvement	Stabilization or worsening

VCmax—maximal vital capacity, TL,co—transfer factor of the lung for carbon monoxide.

**Table 2 diagnostics-13-01109-t002:** Baseline characteristics of the study group.

Variable	Whole Group *N* = 91
Age at diagnosis (y), mean (±SD)	51.6 (±10.32)
Males, Nº (%)	44 (48)
Ever smokers, Nº (%)	37 (40.7)
VC max (L), mean (±SD)	2.97 (±0.97)
VC max (% pred.), mean (±SD)	82.4 (±20.1)
TLC (L), mean (±SD)	5.32 (±1.15)
TLC (% pred.), mean (±SD)	94.7 (±21.7)
RV%TLC (% pred.), mean (±SD)	114.2 (±21.86)
TLco (% pred.), mean (±SD)	48.87 (±14.9)
Tiffenau index (%), mean (±SD)	78 (±10.9)
6MWD (m), mean (±SD)	480.1 (±106.2)
6MWT SpO_2_-1 (%), mean (±SD)	95.4 (±2.3)
6MWT SpO_2_-2 (%), mean (±SD)	87.8 (±7.3)
6MWT desaturation (%), mean (±SD)	7.82 (±6.67)
Distance/desaturation index (DDI), mean (±SD)	155.1 (±172.9)
Distance-saturation product (DSP), mean (±SD)	424.2 (±110.4)
Fibrotic HP Nº (%)	52 (57)
Non-fibrotic HP Nº (%)	39 (43)
Prednisone monotherapy Nº (%)	76 (83.5)
Prednisone + azathioprine Nº (%)	15 (16.5)

VC max—maximal vital capacity, TLC—total lung capacity, RV—residual volume, TL,co—transfer factor of the lungs for carbon monoxide, 6MWD—6 min walk distance, 6MWT—6 min walk test, 6MWT SpO_2_-1—6MWT oxygen saturation before the test, 6MWT SpO2-2—6MWT oxygen saturation after the test.

**Table 3 diagnostics-13-01109-t003:** Comparison of baseline 6MWT parameters (mean ± SD) in fibrotic versus non-fibrotic HP (Mann-Whitney test).

Parameter	Whole Group	fHP	Non-fHP	*p*
6MWD (m)	485.7 (±106.3)	472.9 (±116.4)	484.0 (±98.48)	0.79
6MWT SpO_2_-1 (%)	95.38 (±2.72)	95.67 (±2.03)	95.03 (±2.54)	0.28
6MWT SpO_2_-2 (%)	87.59 (±7.38)	87.19 (±7.39)	88.64 (±7.21)	0.23
6MWT SpO_2_ desaturation (%)	7.82 (±6.67)	8.48 (±6.554)	6.39 (±6.548)	0.05
DDI	155.1 (±172.9)	134.4 (±139.8)	182.8 (±175.3)	0.07
DSP	424.2 (±110.4)	425 (±103.7)	423.2 (±120.3)	0.83

6MWD—6 min walk distance, 6MWT—6 min walk test, 6MWT SpO_2_-1—6MWT oxygen saturation before the test, 6MWT SpO_2_-2—6MWT oxygen saturation after the test, DDI—distance/desaturation index, DSP—distance*saturation product.

**Table 4 diagnostics-13-01109-t004:** Correlation between baseline 6MWT and PFT parameters (Spearman).

Parameter	VC max %pred			TL,co %pred		
	r	95% CI	*p*	r	95% CI	*p*
6MWD	−0.0206	−0.1897 to 0.2292	0.8444	0.3616	0.1645 to 0.5309	0.0004
6MWT SpO_2_ desaturation (%)	−0.4463	−0.5997 to −0.2611	<0.0001	−0.6592	−0.7633 to −0.5217	<0.0001
DSP	0.1309	−0.0809 to 0.3314	0.2111	0.5134	0.3405 to 0.6528	<0.0001
DDI	0.3890	0.1953 to 0.5534	0.0001	0.6525	0.5131 to 0.7583	<0.0001

VC max %pred—maximal vital capacity % of predicted, TL,co %pred—transfer factor of the lungs for carbon monoxide % of predicted, 6MWD—6 min walk distance, 6MWT—6 min walk test, DSP—distance*desaturation product, DDI—distance/desaturation index.

**Table 5 diagnostics-13-01109-t005:** 6MWT results (medians with 95% CI) before and after treatment according to treatment response (Wilcoxon test).

Parameter	Whole Group (*n* = 91)		*p*	Responders (*n* = 63)		*p*	Non-Responders (*n* = 28)		*p*
	Before Treatment	After Treatment		Before Treatment	After Treatment		Before Treatment	After Treatment	
6MWD (m)	479 (458–502)	524 (489–536)	0.0001	479 (453–503)	524 (475.5–534)	0.0036	479.5 (437–531)	525.5 (491.5–568)	0.0112
6MWT SpO_2_ 1 (%)	96 (94.9–95.9)	96 (95.8–96.6)	0.0017	96 (95–96)	96 (96–96.9)	0.0015	96 (94.1–96.1)	96 (94.6–96.5)	0.3347
6MWT SpO_2_ 2 (%)	89 (86.3–89.3)	92 (88.8–91.5)	0.0171	89 (86–89.5)	92 (88.8–92)	0.0015	88.5 (84.7–91.2)	92 (87–92.1)	0.1487
6MWT desat (%)	6 (6.23–8.96)	4 (4.89–7.24)	0.0073	7 (6.3–9.3)	4 (4.7–7.5)	0.0093	5 (4.2–10.2)	4 (3.8–8.1)	0.4797
DSP	416.5 (401.2–447.2)	471.6 (440.3–487.0)	<0.0001	414.7 (395.8–446.9)	471.6 (428.8–488.8)	0.0004	421.6 (380.1–481.3)	470.7 (437.6–511.7)	0.0118
DDI	70.7 (119.1–191.1)	117.7 (155.7–231.9)	0.0765	65.7 (91.8–165)	144.0 (167.7–264.7)	0.0019	111.0 (132.3–297.8)	81.9 (85.1–201.5)	0.2016

6MWD—6 min walk distance, 6MWT—6 min walk test, 6MWT SpO2-1—6MWT oxygen saturation before the test, 6MWT SpO2-2—6MWT oxygen saturation after the test, DSP—distance*saturation product, DDI—distance/desaturation index.

**Table 6 diagnostics-13-01109-t006:** Treatment-related change (medians with 95% CI) in 6MWT parameters according to the type of response to treatment (Mann-Whitney test).

Parameter	Treatment Related Changes, Median (95% CI)		*p*
	Responders	Non-Responders	
6MWT desat. change (%)	−2 (−4.08–−1.61)	−1.5 (−2.98–0.12)	0.3475
6MWT dist change (m)	51 (36–72)	10.5 (−61.2–27.9)	0.0056
DSP post/DSP pre	1.09 (1.04–1.19)	1.09 (1.06–1.24)	0.6428
DDI post/DDI pre	1.67 (1.85–3.63)	0.88 (0.7–1.73)	0.0001

6MWT—6 min walk test, DSP—distance*saturation product, DDI—distance/desaturation index.

## Data Availability

The data are available from the corresponding author upon request.

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
