# Peer review of "New 6-Minute-Walking Test Parameter—Distance/Desaturation Index (DDI) Correctly Diagnoses Short-Term Response to Immunomodulatory Therapy in Hypersensitivity Pneumonitis"

_diagnostics, 2023, doi:10.3390/diagnostics13061109_

Round 1
Reviewer 1 Report
This a nice study that connects treatment response with exercise capacity in HP patients. The results are well presented and the manuscript is well written.
It would be useful to be specified how the patients were divided to responders and non responders (eg were there any cut offs in PFT?).
What about patients that remained stable?
Reviewer 2 Report
1. The wrong use of paragraphs all over the manuscript should be addressed.
2. Materials and Methods should be elaborated and explained further
3. The discussions should be supported with tables for understanding and smooth flow of the manuscript
Round 2
Reviewer 2 Report
The authors have addressed some of my earlier concerns however i still have issues on the presentation of the paragraphs in introductory section. join the paragraphs with less than 6 lines and arrange the flow. minimize the use of paragraph on every sentence.
The conclusion is very bore and must be ellaborated to be precise but rich in content as the summary of the work done, Also limitation and future work of the proposed outlined should be highlighted for future purposes.
